# A *Plasmodium yoelii* HECT-like E3 ubiquitin ligase regulates parasite growth and virulence

Sethu C. Nair[1], Ruixue Xu[2], Sittiporn Pattaradilokrat[1,3], Jian Wu[1], Yanwei Qi[1,2], Martine Zilversmit[1], Sundar Ganesan[4], Vijayaraj Nagarajan[5], Richard T. Eastman[1], Marlene S. Orandle[6], John C. Tan[7], Timothy G. Myers[8], Shengfa Liu[2], Carole A. Long[1], Jian Li[2] & Xin-zhuan Su[1,2]

Infection of mice with strains of *Plasmodium yoelii* parasites can result in different pathology, but molecular mechanisms to explain this variation are unclear. Here we show that a *P. yoelii* gene encoding a HECT-like E3 ubiquitin ligase (*Pyheul*) influences parasitemia and host mortality. We genetically cross two lethal parasites with distinct disease phenotypes, and identify 43 genetically diverse progeny by typing with microsatellites and 9230 single-nucleotide polymorphisms. A genome-wide quantitative trait loci scan links parasite growth and host mortality to two major loci on chromosomes 1 and 7 with LOD (logarithm of the odds) scores = 6.1 and 8.1, respectively. Allelic exchange of partial sequences of *Pyheul* in the chromosome 7 locus and modification of the gene expression alter parasite growth and host mortality. This study identifies a gene that may have a function in parasite growth, virulence, and host–parasite interaction, and therefore could be a target for drug or vaccine development.

[1] Malaria Functional Genomics Section, Laboratory of Malaria and Vector Research, National Institute of Allergy and Infectious Disease, National Institutes of Health, Bethesda, MD 20892, USA. [2] State Key Laboratory of Cellular Stress Biology, Innovation Center for Cell Signaling Network, School of Life Sciences, Xiamen University, Xiamen, Fujian 361005, China. [3] Department of Biology, Faculty of Science, Chulalongkorn University, Bangkok 10330, Thailand. [4] Biological Imaging Section, Research Technology Branch, National Institute of Allergy and Infectious Disease, National Institutes of Health, Bethesda, MD 20892, USA. [5] Bioinformatics and Computational Biosciences Branch, Office of Cyber Infrastructure and Computational Biology, National Institute of Allergy and Infectious Disease, National Institutes of Health, Bethesda, MD 20892, USA. [6] Comparative Medicine Branch, National Institute of Allergy and Infectious Diseases, National Institutes of Health, Bethesda, MD 20892, USA. [7] The Eck Institute of Global Health, Department of Biological Sciences, University of Notre Dame, Indiana 46556, USA. [8] Genomic Technologies Section, Research Technologies Branch, National Institute of Allergy and Infectious Disease, National Institutes of Health, Bethesda, MD 20892, USA. Sethu C. Nair and Ruixue Xu contributed equally to this work. Correspondence and requests for materials should be addressed to J.L. (email: jianli_204@xmu.edu.cn) or to X.-z.S. (email: xsu@niaid.nih.gov)

Malaria is a fatal disease responsible for ~ 429,000 deaths globally in 2015[1]. Clinical symptoms are manifested during the erythrocytic stages that have been studied extensively to understand host immune responses and molecular mechanisms of the disease. As a result of factors that are difficult to control in clinical settings, such as variation in host and parasite genetic backgrounds, mixed-strain infections, co-infection with other pathogens, and the timing of infection, a number of mouse malaria models have been developed for studying host responses and for testing candidate antimalarial drugs and vaccines. The use of inbred mice and cloned malaria parasites enables control of host and parasite genetic backgrounds, and infections of mice can be performed in a controlled environment, although the results obtained from studies of rodent malaria parasites need to be verified by data from human malaria infections.

The genetic backgrounds of both host and parasite can greatly influence malaria disease severity. A broad spectrum of host responses, including different parasite growth patterns and host mortality, was detected when 25 inbred mouse strains were infected with *Plasmodium chabaudi chabaudi* AS parasite[2]. On the other hand, infections with different parasite strains, including those with almost identical genomes (isogenic), often lead to different disease phenotypes. For example, BALB/c mice infected with the parasite *Plasmodium yoelii yoelii* 17XL (or YM) die within 7 days of infection (p.i.), whereas mice infected with its isogenic strain *P. y. yoelii* 17XNL recover from infection[3, 4]. Infection with isogenic parasites *P. y. nigeriensis* N67 (N67) and *P. y. nigeriensis* N67C (N67C) also results in differences in parasitemia, tissue pathology, and host mortality[5]. C57BL/6 mice infected with N67C die 7 days after inoculation with $1 \times 10^6$ infected red blood cells (iRBC), whereas mice infected with N67 survive for more than 15 days with parasitemia declining to below 5–10% on day 7. Similarly, the isogenic parasites *P. berghei* ANKA and *P. berghei* NK65 produce different disease phenotypes; *P. b.* ANKA infection leads to neurologic symptoms and is considered a model of human cerebral malaria, whereas mice infected with *P. b.* NK65 do not have any neurological symptoms[6, 7]. These studies demonstrate that both host and parasite genetics are important in disease manifestation, and these variations provide important disease models for dissecting the molecular mechanisms underlying disease phenotypes.

Host immune responses are critical for the control of parasite multiplication and disease severity[8, 9]. Macrophage-mediated killing seems to be important in the early stages of *P. yoelii* infection in mice[10, 11]. At a molecular level, Toll-like receptors, CD36, Nalp3, and other molecules have been shown to have an important function in host innate responses against malaria parasites through the recognition of parasite-specific molecules[9, 12–15]. Type 1 interferon responses mediated by

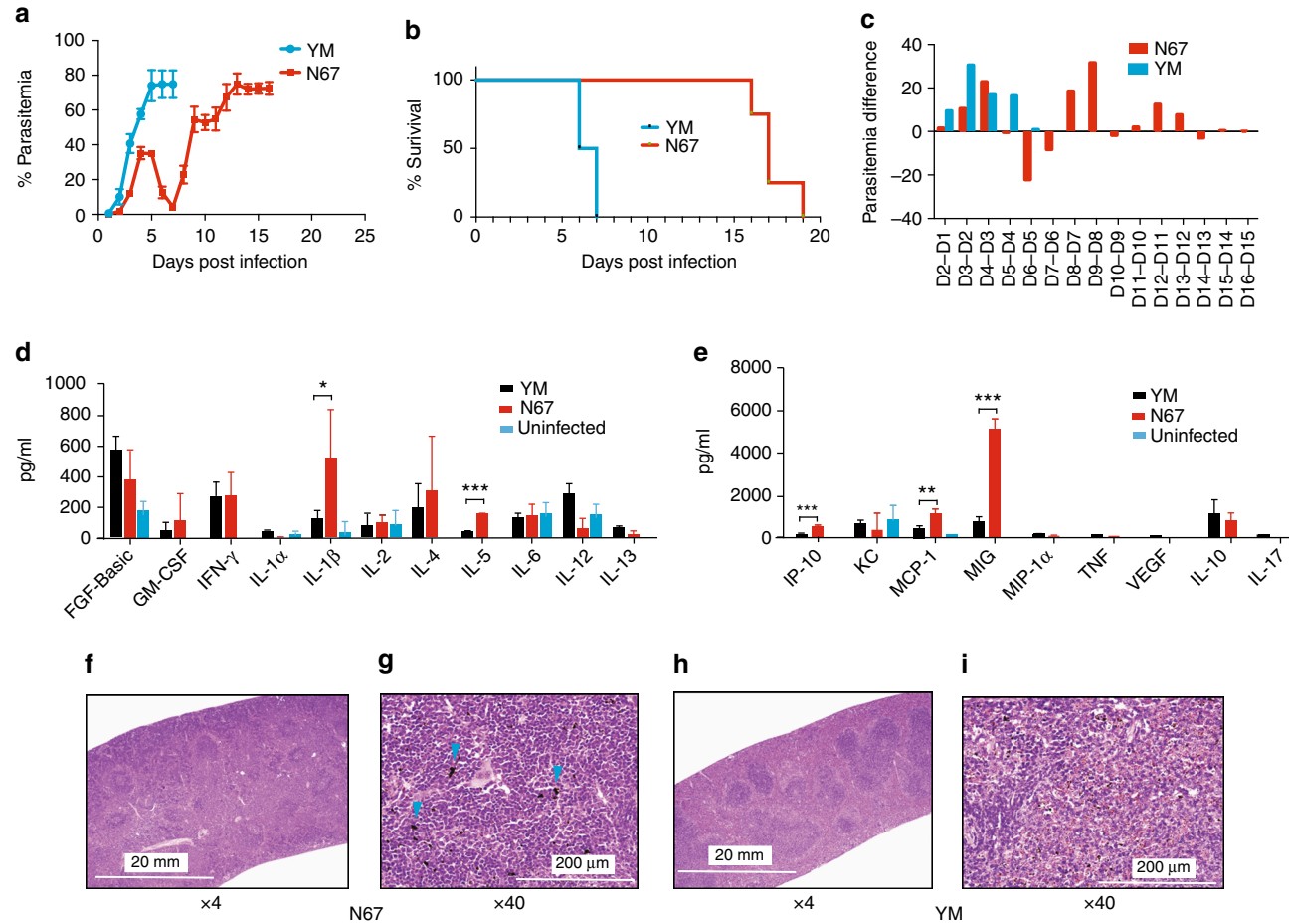

**Fig. 1** Disease phenotypes in mice infected with *P. y. yoelii* YM and *P. y. nigeriensis* N67. **a** Daily mean parasitemia in mice infected with YM or N67 parasite. **b** Mortality rates of mice infected with YM or N67. **c** Differential parasitemia of mice infected with YM or N67 (parasitemia minus that of previous day). **d**, **e** Levels of cytokines and chemokines in mouse plasma day-4 post infection with YM or N67. For (**a**, **b**, **d**, **e**) means and SD's were obtained from five mice each. Two-sided *t*-test; *P < 0.05, **P < 0.01, ***P < 0.001. **f–i** Hematoxylin and eosin (H&E) stain of mouse spleens on day 4 post infection with YM or N67 (at ×4 or ×40 magnification). Note splenomegaly in N67-infected mice. The *blue arrowheads* point to large hemozoin pigments

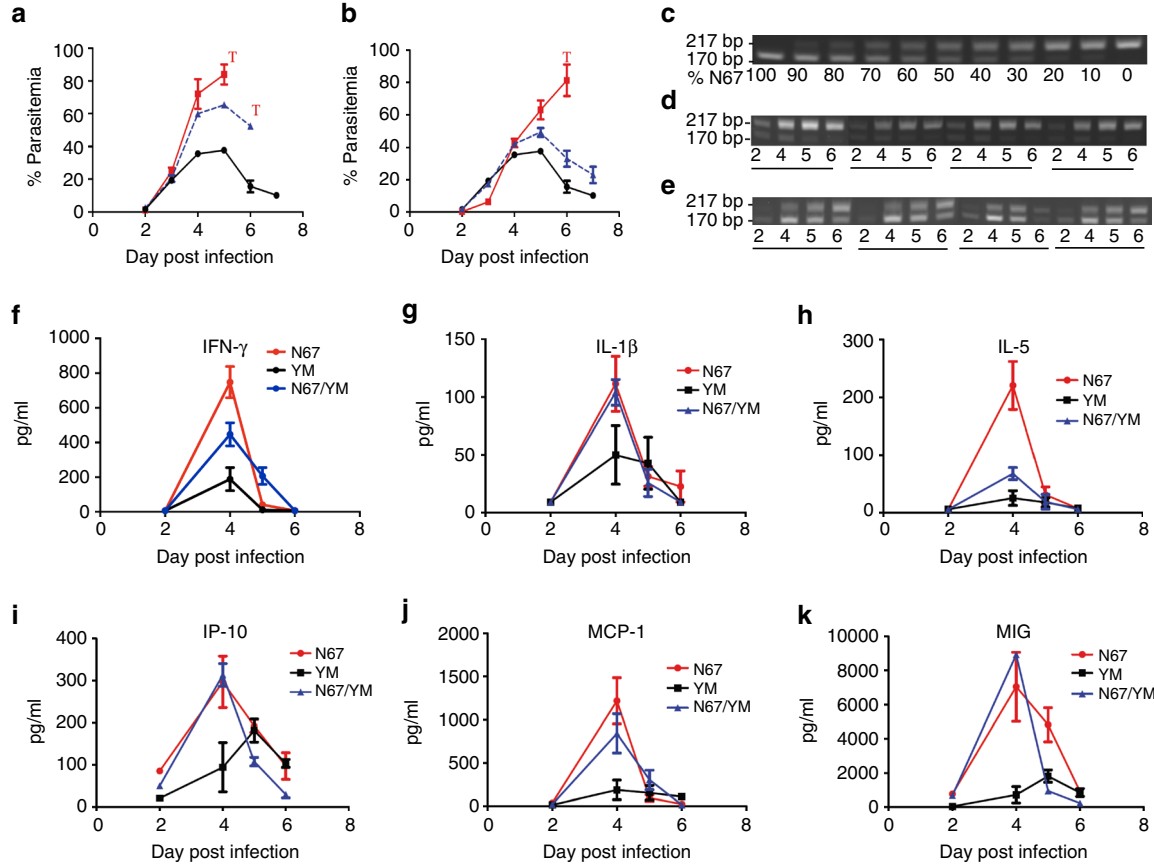

**Fig. 2** Parasitemia and cytokine responses from mice co-infected with N67 and YM parasites. **a** Parasitemia from mice infected with 1:1 ratio of YM and N67. *Red line*, mice infected with $5 \times 10^5$ YM; *black line*, infected with $5 \times 10^5$ N67; *blue line*, infected with a mixture of $5 \times 10^5$ YM and $5 \times 10^5$ N67. 'T' indicates termination of experiments due to host death or moribund (and were humanely killed). **b** The same experiments as in (**a**) except $5 \times 10^4$ YM and $5 \times 10^5$ N67 parasites were used in either individual or mixed infections. **c** PCR products of microsatellite marker PY845-2 from known ratios of N67 and YM DNA. Band intensity for the two alleles shifted as the proportion of N67 changed, suggesting that band intensity could approximate the ratios of the DNAs from the parasites. **d**, **e** Proportional quantifications of YM and N67 DNAs in samples from four mice on days 2–6 infected with inoculum containing the parasite mixture at ratios of 1:1 and 1:10 (YM:N67), respectively. PCR product from N67 (170 bp) and YM (217 bp) are as indicated. **f–k** Plasma cytokine/chemokine profiles from the mixed infection at ratio of 1:10 (YM:N67) in (**b**). For (**a**, **b** and **f–k**) means and SD bars were from four to five mice each

cytosolic sensor MDA-5 have been shown to be activated during *P. yoelii* and *P. berghei* infections, resulting in the control of parasitemia during early phases of infection[5, 16, 17]. However, high levels of type 1 and 2 interferons may contribute to severe disease and host death, given that *Ifnar1*[−/−] mice are more resistant to experimental cerebral malaria after *P. berghei* ANKA infection[18–20]. Many other parasite molecules and proteins have been shown to modulate host immune responses through interactions with specific host receptors, including proteins, nucleic acids, and lipids[12, 21–26].

To investigate the molecular basis of malaria pathogenesis, here we genetically cross two lethal *P. yoelii* parasites, N67 that kills the host ~15 days p.i., and YM that kills the host in 7 days, probably due to destruction of RBCs. We obtain 43 independent recombinant progenies (IRPs) after genotyping cloned progeny using microsatellites and a custom-made single-nucleotide polymorphism (SNP)-typing microarray[27]. Quantitative trait loci (QTL) analysis of the progeny genotypes and disease phenotypes links two loci on chromosome 1 (Chr1) and chromosome 7 (Chr7) to control of parasitemia and/or host mortality. Genetic modifications of a putative HECT-like E3 ubiquitin ligase gene (*Pyheul*) with the highest LOD score in the Chr7 locus alters the gene transcript level, parasite growth, and host mortality, suggesting a role for this gene in host–parasite interactions.

## Results

**Different parasite growth rates and host responses**. Different strains of *P. yoelii* have very distinct growth and virulence phenotypes in mice[28]. To identify parasite genes playing a role in virulence and disease severity, we first evaluated variations in disease phenotype after infection with two lethal parasites, YM and N67. Intravenous injection of C57BL/6 with $1 \times 10^6$ iRBCs of YM strain led to host death in 6–7 days, largely due to high parasitemia (70–80%) and rapid lysis of iRBCs (Fig. 1a, b). In contrast, the parasitemia in N67-infected mice reached 30–40% in 4 days, and declined to under 5% after 7 days. The parasitemia rose again to 60–70%, leading to host death on day 15–20 (Fig. 1a, b). The ability of the mice infected with N67 to control early parasitemia could be quantified by subtracting the parasitemia measurements from that of the previous day, showing a clear dip in parasitemia between 5 and 6 days of infection (Fig. 1c). Comparison of serum levels of 20 cytokines/chemokines in mice infected with YM and N67 strains between days 2 and 16 p.i. showed significantly higher pro-inflammatory cytokine/chemokine responses in N67-infected mice on day 4 p.i., including interleukin (IL)-5, IP-10, MIG, MCP-1, and TNF, but higher levels of FGF-basic (basic fibroblast growth factor), GM-CSF, KC, IL-1α, IL-13, IL-17, and VEGF in YM-infected mice on day 2 and/or day 3 day p.i. (Fig. 1d, e and Supplementary Fig. 1).

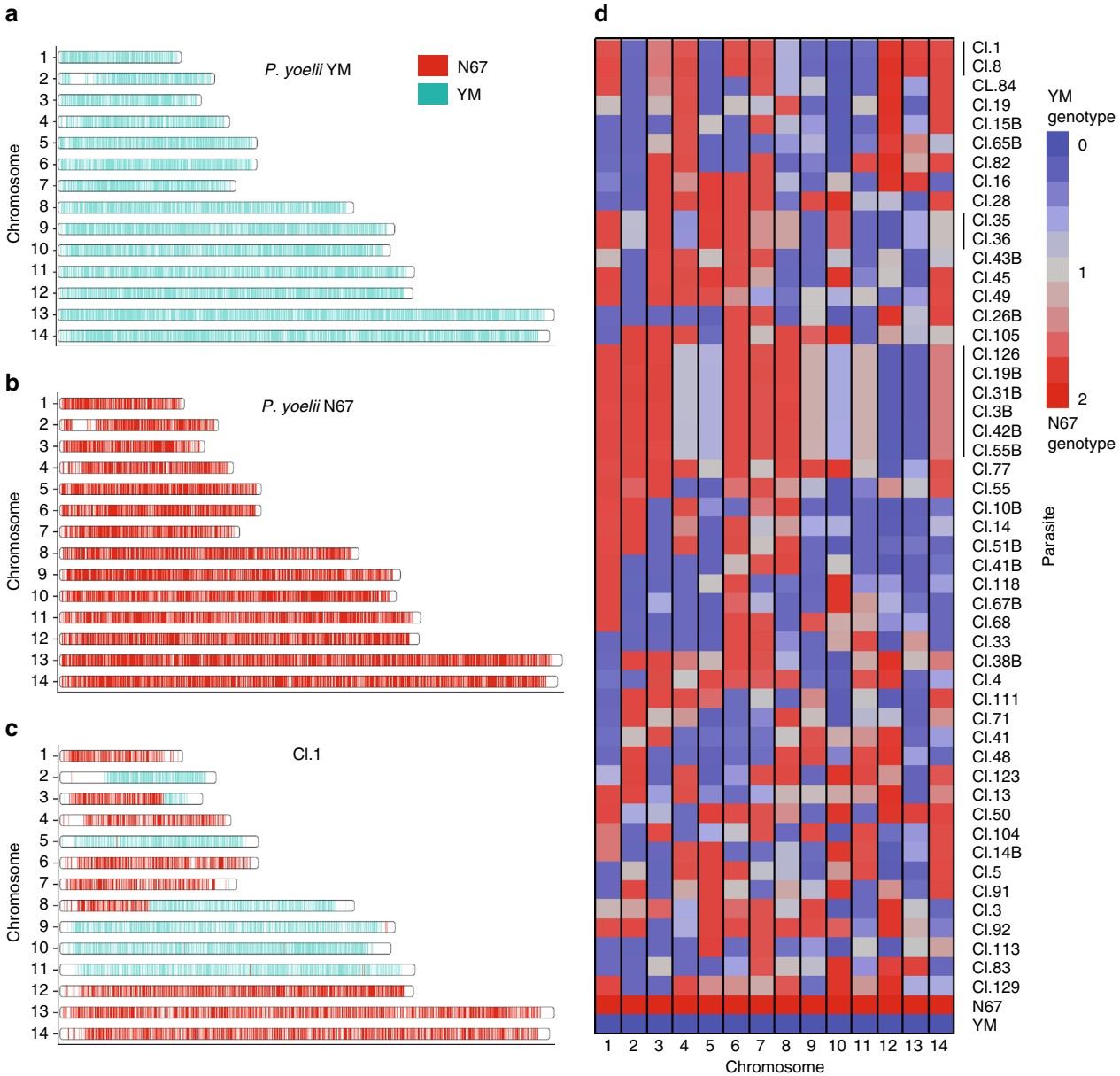

**Fig. 3** Genome-wide genotypes of progeny from the YM × N67 cross. **a–c** Ideograms of single-nucleotide polymorphisms (SNPs) distributed across the parasite's 14 chromosomes. Progeny Cl.1 was chosen for an arbitrary example of observed genetic cross. *Cyan*, YM genotype; *red*, N67. **d** Patterns of inheritance for 43 progenies and their parents based on per-chromosome average of numerically encoded genotypes (0 = YM genotype, 2 = N67 genotype, 1 = missing genotype call; see Methods). The mean genotype score for the of 14 chromosomes from each progeny are indicated in the heatmap. *Blue* indicates a chromosome with predominatly YM genotype; *red*, predominantly N67 genotype; *gray*, equal genotype representation

Histochemical examination of spleens of infected animals 4 days p.i. revealed splenomegaly with large hemozoin pigments (*blue arrowheads*) and mild/moderate lymphoid hyperplasia in N67-infected mice, suggesting infiltration of macrophages and active phagocytosis of iRBCs (Fig. 1f, g). In the YM-infected mice, there was mild lymphoid necrosis and increased numbers of hemozoin-laden macrophages and iRBCs between host cells, indicating low levels of phagocytic activities (Fig. 1h, i and Supplementary Table 1).

**Inhibition of YM parasitemia by co-infection with N67.** The YM parasite (or 17XL) is a fast-growing parasite derived from the nonlethal, slow-growing parasite 17X[29]. The fast-growing phenotype has been attributed to its ability to invade both young and mature RBCs efficiently[3, 29]. However, it is also

possible that the YM parasite can grow rapidly because it has mechanisms to evade or suppress host immune responses. To test the hypothesis that the immune response associated with N67 infection is critical in controlling parasite growth, we co-infected mice with YM and N67 at ratios of 1:1 and 1:10 (YM: N67; Fig. 2a, b), respectively. In mice infected with a 1:1 mixture of YM and N67 ($5 \times 10^5$ YM plus $5 \times 10^5$ N67), the parasitemia peaked on day 5 *p.i.* and then declined (as N67 did, also infected with $5 \times 10^5$), whereas parasitemia from mice infected with YM alone ($5 \times 10^5$) was higher than that of the mixed infections and continued to increase on day 6 (Fig. 2a). Similar results were obtained in infections with 1:10 ratio (YM: N67), with lower parasitemia in mice infected with the mixed population than those, given the 1:1 ratio (Fig. 2b). Microsatellite (PY845-2) analysis of amplified DNA samples from the mixed infections showed that YM (*top band*) quickly overgrew N67 from days 4 to

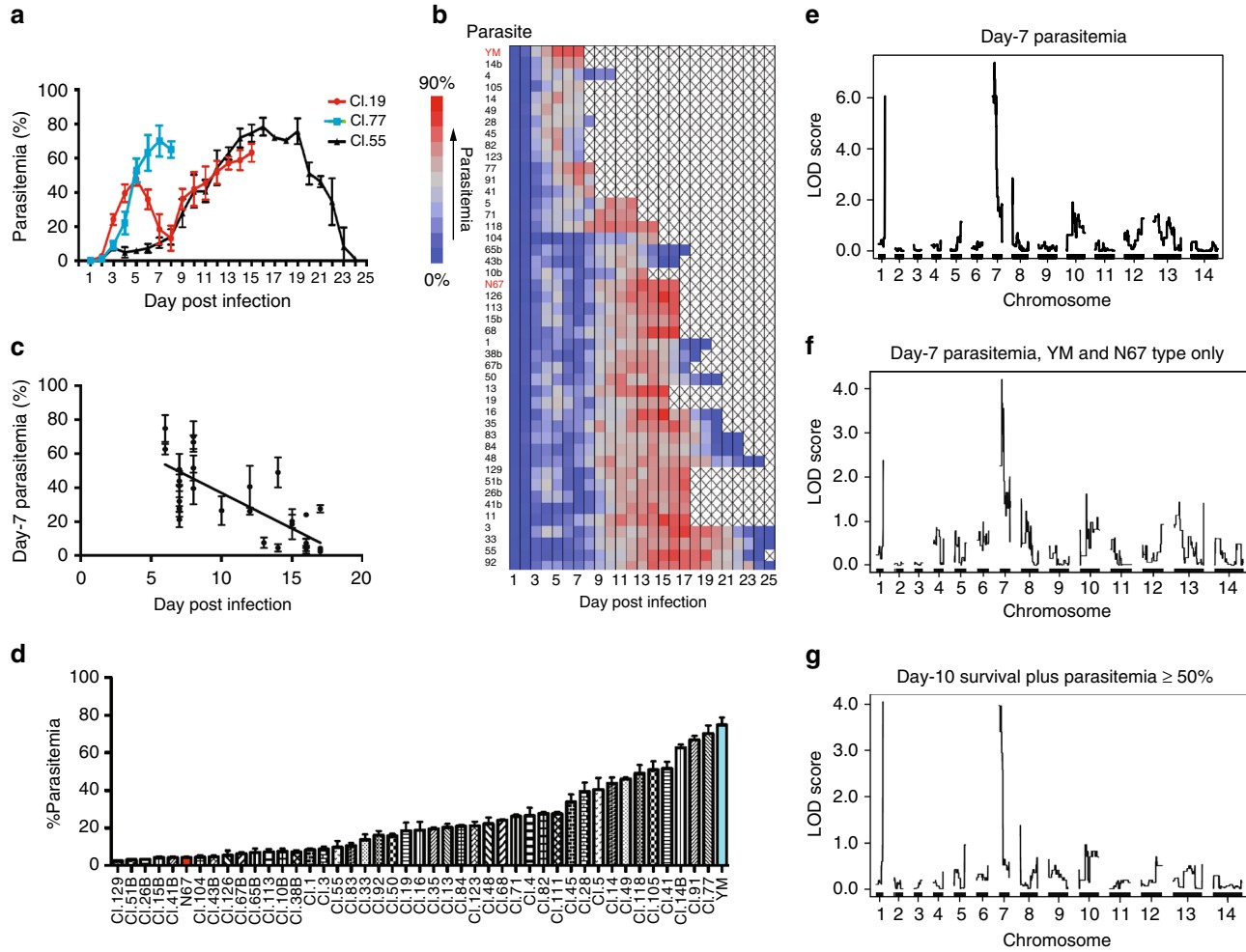

**Fig. 4** Parasitemia of YM × N67 cross progeny and LOD scores from quantitative trait loci analysis (QTL). **a** Parasitemia plots from three exemplar progeny. Mean parasitemia and SDs were from four mice. The parasitemia time course of Cl.77 is similar to that of YM; parasitemia of Cl.19 is similar to that N67; parasitemia of Cl.55 is similar to that of N67, but also declines to zero on day 25, representing a new phenotype. **b** Parasitemia time course of the two parents and 43 progenies arranged in order of pattern similarity. Missing (post mortem) measurements are indicated by "X" in the heatmap. **c** Correlation plot of day-7 parasitemia and the date of host death. Note that some of the progenies did not kill the host and were not plotted. **d** The mean day-7 parasitemia with SE's of four mice from the parents (YM and N67) and their 43 progenies. Progeny names are as labeled. **e** LOD (logarithm of the odds) score plot of QTL analysis of day-7 parasitemia and genome-wide SNP genotypes of the progeny. **f** LOD score plot of QTL analysis of day-7 categorical phenotypes from 36 progenies with parasitemia patterns similar to either YM or N67 (excluding progeny Cl.4, Cl.14, Cl.28, Cl.45, Cl.49, Cl.82, and Cl.91 that died with declining parasitemia or intermediate phenotypes). **g** LOD score plot of QTL analysis of day-10 host survival plus parasitemia ≥50% when it died as phenotypes. QTL analysis was performed as described[28, 58]

6 in the 1:1 infection; moreover, the majority of the parasite populations had the YM genotype on days 4–6 (Fig. 2c, d). If the parasite growth was controlled purely by invasion efficiency, we would have seen a continuing increase of parasitemia in the mice with mixed infection on day 6 p.i, not a decline, because the parasites were effectively YM populations (as seen in mice infected with $5 \times 10^5$). The parasitemia suppression was even more obvious with the mixed infection 1:10 ratio ($5 \times 10^4$ YM plus $5 \times 10^5$ N67). Similar to those infected with $5 \times 10^5$ N67, the parasitemia in mice infected with mixed parasite populations dramatically decreased from day 5, even though the ratios of the parasites were ~1:1 (Fig. 2e). The parasitemia of the mixed-infected mice were ~30% or about twice that of N67-infected mice, although the parasitemia in the YM-infected mice was higher than 80% on day 6. The results suggested that declines of parasitemia in the mixed-infected mice after day 5 were likely due to inhibition of YM parasite growth by host immune responses, due largely to N67 co-infection. Indeed, the levels of INF-γ, IL-1β, IL-5, IP-10, MCP-1, and MIG were elevated in the mice given the

mixed-infection at a 1:10 ratio compared with those infected with YM alone (Fig. 2f, k). These results show that the host mounted a stronger response to N67 than to YM infection despite higher YM parasitemia, and the response to N67 could influence the growth and disease outcome of the fast-growing YM parasite. The results also suggest that the ability to induce protective immune responses is strain-dependent (N67-specific), but the protective responses are strain-transcendent if properly induced. Identification of the elements of N67 parasite stimulating this strong cross-strain response is critical for understanding the mechanism of disease pathogenesis.

**Genetic crosses and genotyping**. To investigate the parasite factors that may contribute to immune responses resulting in the decline of parasitemia and delayed host mortality in the N67-infected mice, we genetically crossed the YM and N67 parasites using procedures described previously[28, 30]. The parasites were initially tested for their infectivity in mosquitoes,

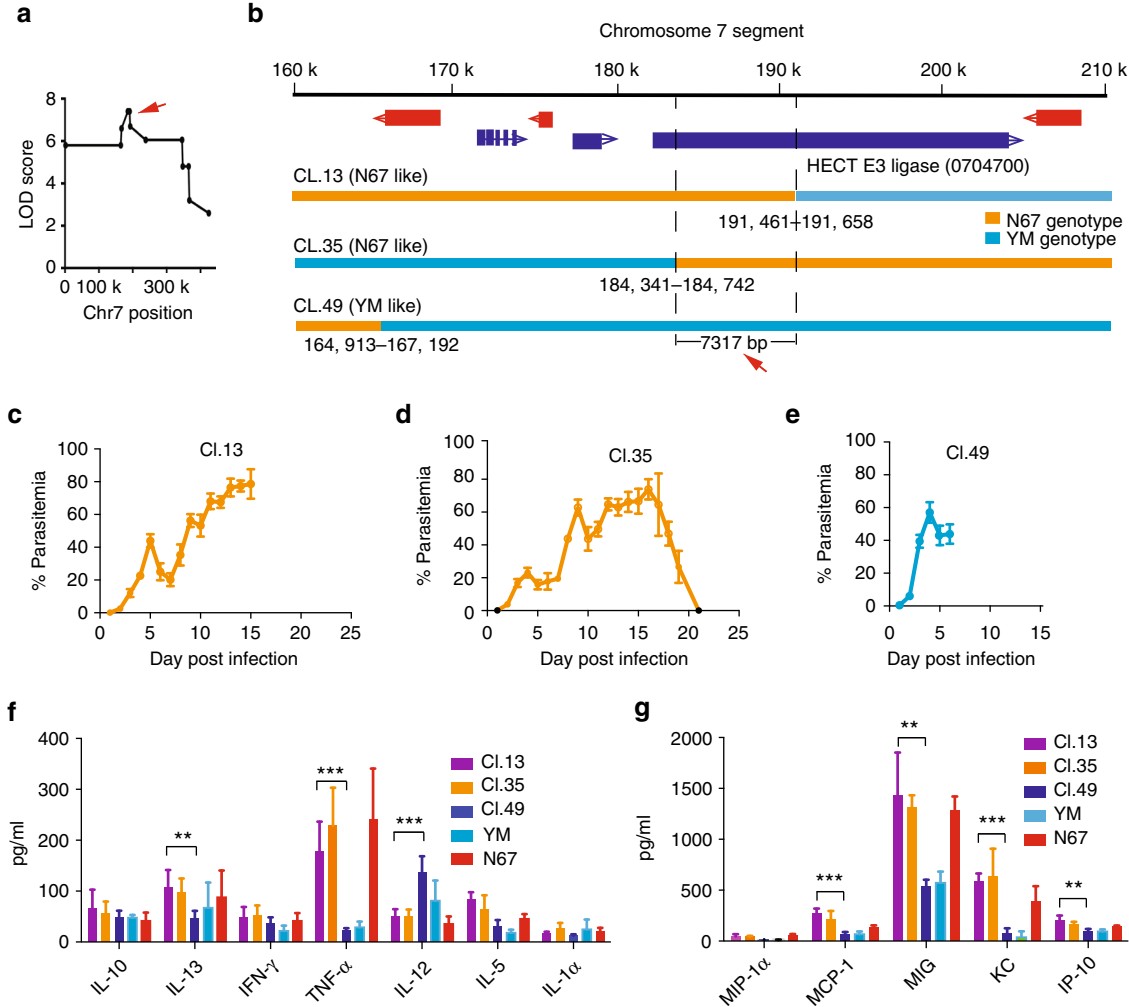

**Fig. 5** Fine-mapping of the chromosome 7 locus linked to parasitemia and host mortality. **a** Plot of peak LOD scores at the chromosome 7 locus, showing the site with peak LOD score (*red arrow*). **b** Expanded view of the chromosome 7 locus showing candidate genes and crossover sites for three progenies with genetic recombination within the locus. Genetic markers in a 7.3 kb segment has the highest LOD score defined by crossovers in progenies Cl.13 and Cl.35. **c–e** Plots of the mean parasitemia with SD from three to five mice infected with three progeny as indicated. **f, g** Cytokine/chemokine levels (means and SDs) from five mice infected with the parents and three progenies, measured using a mouse magnetic 20-plex kit (Invitrogen) in a Luminex-200 machine according to the manufacturer's instructions. Two-sided *t*-test, **$P < 0.01$; ***$P < 0.001$

and differences in oocyst counts were observed. N67 produced ~ 10–12-fold more viable oocysts than the YM parasite (Supplementary Fig. 2), representing another interesting phenotypic difference that can be studied later. The ratio of midgut oocyst counts was shown to influence the efficiency in generating an optimum number of recombinant progeny in a genetic cross[30, 31]; therefore, we used a parasite ratio of YM:N67 (9:1) in our crossing experiments following the procedures described previously[32]. Recombinant progeny was cloned from a parasite mixture in the mouse blood after limiting dilution and injection of 100 µl medium containing ~ 1 parasite into a mouse. We performed five independent crossing experiments between YM and N67 parasites and obtained 188 infected mice after injecting 500 mice.

To determine which parasites from the infected mice are recombinant progeny, we genotyped the 188 progenies using microsatellite markers as described previously[28]. To detect the size differences of the PCR products on agarose gels, we first selected 10 markers on 10 of the 14 parasite chromosomes that had PCR products with large differences in size (≥8 bp) between

the two parents (Supplementary Fig. 3 and Supplementary Table 2). Initial screening of the progeny using the microsatellite markers identified 60 progenies that had unique microsatellite marker combinations, suggesting IRP. To obtain genome-wide genotypes of the progeny, we used a custom *P. yoelii* SNP detection microarray to genotype the 60 putative IRPs identified after microsatellite typing. Genomic DNAs from individual progeny were hybridized to the array, and signals were scanned and analyzed as described previously[27]. The use of high-density marker array detected additional redundant clones. Of the 60 recombinant progenies, four had parental genotypes, six had mixed genotypes, and 50 were recombinant progeny. Among the recombinant progeny, seven were redundant recombinant progeny with the same genotypes and 43 were IRPs (Fig. 3a–d, Supplementary Fig. 4, and Supplementary Data 1). In total, 249 recombination events were detected among the 43 progenies, averaging 5.7 crossovers per parasite per sexual generation, leading to an estimated recombination rate of ~ 40 kb/cM (based on a genome size of 23 Mb). This result is consistent with the previous estimate of 39.7 kb/cM[28].

**Disease phenotypes of the recombinant progeny**. To characterize the growth dynamics of the progeny and disease severity they caused, we measured parasitemia in C57BL/6 mice (four mice each) after intravenous injection of $1 \times 10^6$ iRBCs from the 43 progeny and their parents. Parasitemia was measured daily after counting Giemsa-stained thin smears. There were progenies having phenotypes similar to YM (e.g., Cl.77) with parasitemia > 60% on days 5–6 and/or host death on days 7–9 p.i. and progeny having phenotypes similar to that of N67 (e.g., Cl.19) having parasitemia of ~ 40% on days 4–5, declining to under 5% on day 7, and increasing again leading to host death on around days 15–17 (Fig. 4a, b and Supplementary Fig. 5). Interestingly, many progenies had intermediate phenotypes, with some mice being able to greatly reduce parasitemia but not being able to survive the infection (clones Cl.4 and Cl.28). On the other hand, many mice were able to control parasitemia and even cleared the parasites after day 25 (e.g., Cl.55 and Cl.67B), representing a new nonlethal phenotype generated from two lethal parasites (Supplementary Fig. 5). Clustering the progeny based on daily parasitemia levels clearly showed distinctive phenotypes with variations in parasitemia and host survival time among the progeny, suggesting contributions of multiple genes to the phenotypes (Fig. 4b). High early parasitemia (day 7) appeared to be positively correlated with host death rate (Fig. 4c). Clearly, the phenotypes of parasitemia, host mortality, and host immune response are closely related and are influenced by multiple host and parasite factors.

**Genetic loci linked to control of parasitemia and host death**. Parasitemia on day 7 showed the maximum difference between the parental strains (Fig. 1a), and for convenience, we began with QTL analysis[28, 30] using day 7 parasitemia (Fig. 4d). We significantly linked two major loci on Chr1 and Chr7 with LOD = 6.1 and 8.1, respectively, to day 7 parasitemia (a third locus on Chr8 had a LOD = 2.8; Fig. 4e). QTL analysis using categorical phenotypes by classifying 36 progenies with parasitemia patterns similar to either YM or N67 also showed significant linkage (LOD score = 3.8) to the Chr7 locus (Fig. 4f and Supplementary Fig. 5). If we used day-10 host survival plus parasitemia ≥50% when it died as phenotypes, we could detect the same two peaks on Chr1 and Chr7 (Fig. 4g). We also calculated the difference in the parasitemia between day 5 and day 7, which reflects the ability of the host to control parasitemia before day 7, and again linked the phenotype to the same locus on Chr7 (Supplementary Fig. 6a). In addition, we performed QTL analysis on day 5 and day 6 parasitemia, and found that only the Chr7 locus was significantly linked to day 6 parasitemia with LOD = 3.1, although a peak on Chr13 with LOD score = 2.7 was also detected (Supplementary Fig. 6b, c). All these analyses using different parameters suggested significant linkage of a major determinant on the Chr7 locus to the control of parasitemia and possibly host mortality in this particular cross.

**Candidate genes linked to parasitemia and host mortality**. We next used a 1.5 LOD support intervals (markers with LOD scores 1.5 units less than the peak score) to define a mapped locus. We found that the locus on Chr1 was flanked by SNP markers at positions 620325–639363, spanning ~19 kb at the subtelomeric region containing five predicted genes; all of them were genes belonging to multicopy gene families, including two *yir* genes and three fam-a protein genes (Supplementary Table 3). The Chr7 locus spanned ~344 kb (position 1867–345696) containing 95 predicted genes, and the Chr8 covered ~ 27 kb (position 8942–36340) containing seven YIR genes (Supplementary Table 3). To further fine-map the genes in the Chr7 locus, we

identified the markers with peak LOD scores in the locus and progeny with crossovers flanking the markers (Fig. 5a, b). Progeny Cl.13 and Cl.35 had crossovers flanking a DNA segment of ~7.3 kb (position 184,341–191,658) containing markers with the highest LOD scores. Both progenies carried a DNA segment from N67 and had lower day-4 parasitemia and survived longer, similar to those of N67, although Cl.35 was also able to clear the parasites (Fig. 5b–d). Progeny Cl.49 had another crossover ~34 kb away and had genotype and phenotype characteristics similar to those of the YM parent (Fig. 5b, e). The progeny Cl.13 and Cl.35, but not Cl.49, also had significantly higher levels of MIG, IP-10, and MCP-1 as seen in N67-infected mice as well as KC and TNF-α (Fig. 5f, g). These results further link the 7.3 kb DNA segment to differences in host responses and possibly host mortality.

The 7.3 kb DNA segment flanked by the two crossovers in progenies Cl.13 and Cl.35 was a part of the 5′ coding region of a predicted gene encoding a putative HECT E3 ubiquitin ligase (PyHEUL or PYYM_0704700; http://plasmodb.org/plasmo/). This gene has a predicted open reading frame of 22,500 bp (182,666–205165 bp) with ~400 amino acids (AA) at the C terminus having homology to a HEUL-conserved domain. E3 ubiquitin ligases add polyubiquitin chains to proteins targeting the proteins for destruction by proteasomes or activating their targets for signal transduction and other biological processes. Protein database searches did not reveal any other known protein domains reported for the PyHEULs[33], suggesting a unique HEUL with unknown binding specificity. There were two polymorphic microsatellites and 225 SNPs, including 97 nonsynonymous SNPs, between YM and N67; however, no AA substitution was observed in the C-terminal HECT domain[34] (~1.3 kb; Supplementary Table 4). Plotting the nonsynonymous SNP distribution in the gene showed five regions with clusters of SNPs in the 7.3 kb mapped region (Supplementary Fig. 7, red arrows). To determine whether the full 22 kb predicted transcript was expressed or not, we divided the gene into 12 segments and used PCR to amplify the segments from genomic DNA and cDNA. All the segments could be amplified from cDNA, confirming the presence of the entire 22 kb transcript in both YM and N67 parasites, with a potential alternative splicing event in the middle of the ORF (Supplementary Fig. 8 and Supplementary Table 5).

**The *Pyheul* gene is essential for parasite viability**. We next attempted to knockout (KO) the *Pyheul* gene from the parasites using a linear construct to replace the gene with a drug selection cassette and a double crossover strategy[35, 36]. Although we could detect integration of the cassette at the locus transiently using PCR, we were not able to obtain parasites with gene disruption after further drug selection (Supplementary Fig. 9). The experiments were repeated three times with similar results. To further confirm the results that the *Pyheul* is essential for parasite viability, we also used a CRISPR/Cas9 gene-editing method described previously[37] to disrupt the *Pyheul* gene. These experiments were performed in a separate laboratory using N67 and 17XL because YM was not available in the laboratory (Xiamen University, China). 17XL and YM were derived from 17X, and 17XL has the same growth and disease phenotypes as YM[29]. We consider YM and 17XL the same parasite because high day-4 parasitemia and sequencing of the *Pyheul* gene from the parasite (see data below) supported the conclusion. We made two constructs, one to delete the whole-protein-coding region (Supplementary Fig. 10a) and the other to delete a segment in the N-terminal region of the PYHEUL (Supplementary Fig. 11a). Both efforts failed to produce viable parasites with disrupted *Pyheul* gene (Supplementary Figs. 10b and 11b). These results

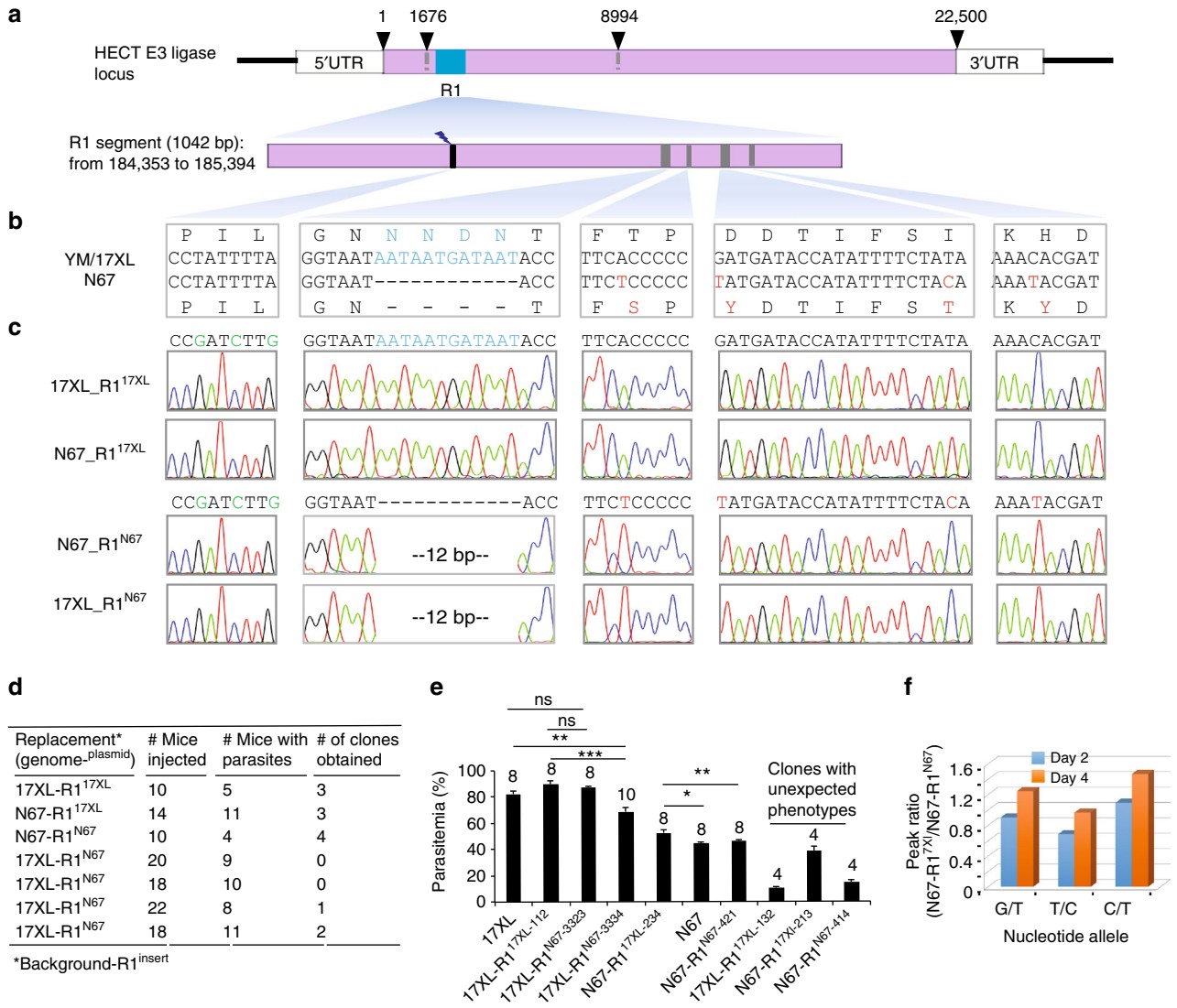

**Fig. 6** Replacement of a partial *Pyheul* gene segment alters parasite growth. **a** Diagram showing *Pyheul* gene structure and the region being replaced using the CRISPR/Cas9 method described previously[37]. The *numbers* on *top* of the gene *bar* are nucleotide positions indicating start and stop positions and the mapped region (1676–8994 bp). **b** Partial nucleotide and amino-acid sequences of guide RNA (gRNA) targeting site (*black*) and the polymorphic sites (*dark green*) between N67 and 17XL parasites within the R1 region. **c** Electropherograms of DNA sequences containing exchanged nucleotides in the self-replacement and cross-replacement parasite clones. Names of recombinant parasite clones are on the *left*. **d** Summary of allelic exchange experiments including the numbers of mice infected and parasite clones obtained. 17XL-R1^N67 indicates R1 segment from N67 parasite replacing that of 17XL, and so on. Numbers of mice injected, numbers of mice injected with transfected parasites; numbers of mice with parasites, numbers of mice with parasites; numbers of clones obtained, numbers of parasite clones with correct modifications obtained. **e** Day-4 parasitemia (means and SE's) of different parasite clones. The *numbers* on *top* of each *bar* indicate numbers of mice used. Two-sided *t*-test, *$P < 0.05$, **$P < 0.01$, ***$P < 0.001$. **f** Ratios of electropherogram peaks at three polymorphic nucleotides between N67 (T-C-T) and 17XI (G-T-C) day 2 (*blue*, 17XL/N67) and day 4 (*orange*) post infection, showing increased proportions of N67-R1^17XL over N67-R1^N67. Parasites were mixed at ~1:1 ratio. The *graph* represents two experiments of similar results

demonstrate that the *Pyheul* gene is an essential gene for parasite viability.

**Replacement of *Pyheul* sequences alters parasite growth.** The YM/17XL parasite grows faster than N67 before day 5 p.i., which could be due to lack of effective immune response and/or high invasion efficiency (Fig. 1a). To investigate whether PyHEUL plays a role in parasite growth, we exchanged a segment within the 7.3 kb mapped region of the *Pyheul* gene between 17XL and N67 parasites. The segment (R1) was a 1042 bp DNA (from184,353 to 185,394 bp) containing four AA substitutions and a NNDN deletion in the N67 parasite (Fig. 6a, b). We

constructed two plasmids each containing the R1 sequence from 17XL or N67, respectively, and transfected each plasmid into both 17XL and N67 for self-replacement and cross-strain replacements after sequencing confirmation of the cloned DNA sequences. After limiting dilution cloning of the parasites grown out of drug selection, we obtained parasites with self-replacement (N67 replacing N67, N67-R1^N67; 17XL replacing 17XL, 17XL-R1^17XL) and allelic exchanged parasites (N67 segment into 17XL genome, 17XL-R1^N67; 17XL segment into N67 genome, N67-R1^17XL; Fig. 6c). Interestingly, clonal N67-R1^N67, 17XL-R1^17XL, and N67-R1^17XL parasites were obtained from a single round of transfection and cloning; however, it was much more difficult to obtain 17XL-R1^N67. Four separate rounds of transfection and cloning

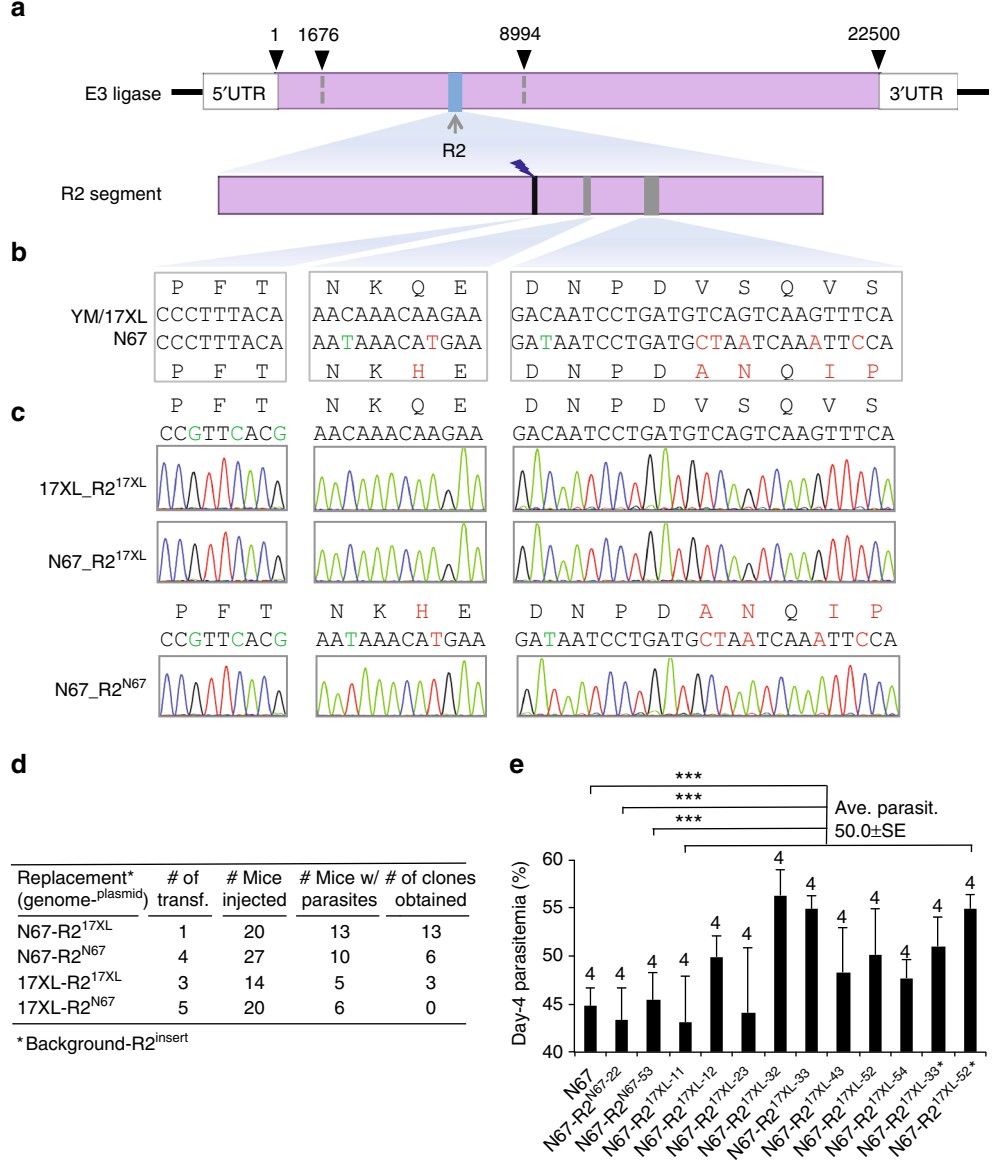

**Fig. 7** Replacement of a second partial *Pyheul* gene segment alters parasite growth. **a** *Diagram* showing *Pyheul* gene structure and the region being replaced using the CRISPR/Cas9 method described previously[37]. The *numbers* on *top* of the gene *bar* are nucleotide positions indicating start and stop positions and the mapped region. **b** Partial nucleotide and amino-acid sequences of guide RNA (gRNA)-targeting site (*black*) and the polymorphic sites (*dark green*) between N67 and 17XL parasites within the R2 region. **c** Electropherograms of DNA sequences containing exchanged nucleotides in the self-replacement and cross-replacement parasite clones. Names of recombinant parasite clones are on the *left*. **d** Summary of allelic exchange experiments including the numbers of transfection experiments (Number of of transf.), numbers of mice injected (Number of mice injected), numbers of mice with parasites (Number of mice w/parasites), and numbers of parasite clones obtained. 17XL-R2$^{N67}$ indicates R2 segment from N67 parasite replacing that of 17XL, and so on. No 17XL-R2$^{N67}$ parasite was obtained after five independent transfection and screenings. **e** Day-4 parasitemia of mice infected with N67, self-replaced (N67-R2$^{N67}$), and 10 clones of allelic exchanged parasites (N67-R2$^{17XL}$), including two repeated clones (N67-R2$^{17XL-33*}$ and N67-R2$^{17XL-52*}$). The means and SEs are from four mice per group (the number of mice in each group is marked on *top* of each *bar*); significance by two-sided *t*-test, ***$P < 0.001$. Averaged day-4 parasitemia from all the N67-R2$^{17XL}$ clones were tested against averaged day-4 parasitemia from N67, N67-R2$^{N67-22}$, and N67-R2$^{N67-53}$. Several groups of the N67-R2$^{17XL}$ clones had a single mouse having parasitemia quite different from the rest of the mice (*outlier*, such as N67-R2$^{17XL-11}$ and N67-R2$^{17XL-23}$) leading to large SE (Supplementary Table 7), which was likely due to technical variation during parasite injection

involving injection of 78 mice were required to finally obtain three 17XL-R1$^{N67}$ parasite clones (Fig. 6d). One possible explanation was that the 17XL-R1$^{N67}$ parasites grew slower than the parental 17XL parasite in the mixture and were overgrown (and missed) when parasite cloning was performed. Infection of mice with the cloned recombinant parasites showed variable results in day 4 parasitemia. We infected BALB/c mice with two 17XL-R1$^{N67}$ clones; one (17XL-R1$^{N67-3334}$) had significantly lower day-4 parasitemia than that of 17XL or 17XL-R1$^{17XL-112}$, but not the

17XL-R1$^{17XL-3323}$ clone (Fig. 6e and Supplementary Table 6). The replacement of N67 R1 segment by that of 17XL significantly increased day-4 parasitemia in the clone N67-R1$^{17Xl-234}$. Unfortunately, we also had some clones that had unexpected growth phenotypes. For example, 17XL-R1$^{17XL-132}$ should grow similarly to 17XL because it was a self-replacement parasite; instead, it had very low day-4 parasitemia (10% day-4 parasitemia). Similarly, N67-R1$^{N67-414}$ and N67-R1$^{17XL-213}$ had parasitemias lower than those of parental parasite N67. For each parasite clone, the day-4

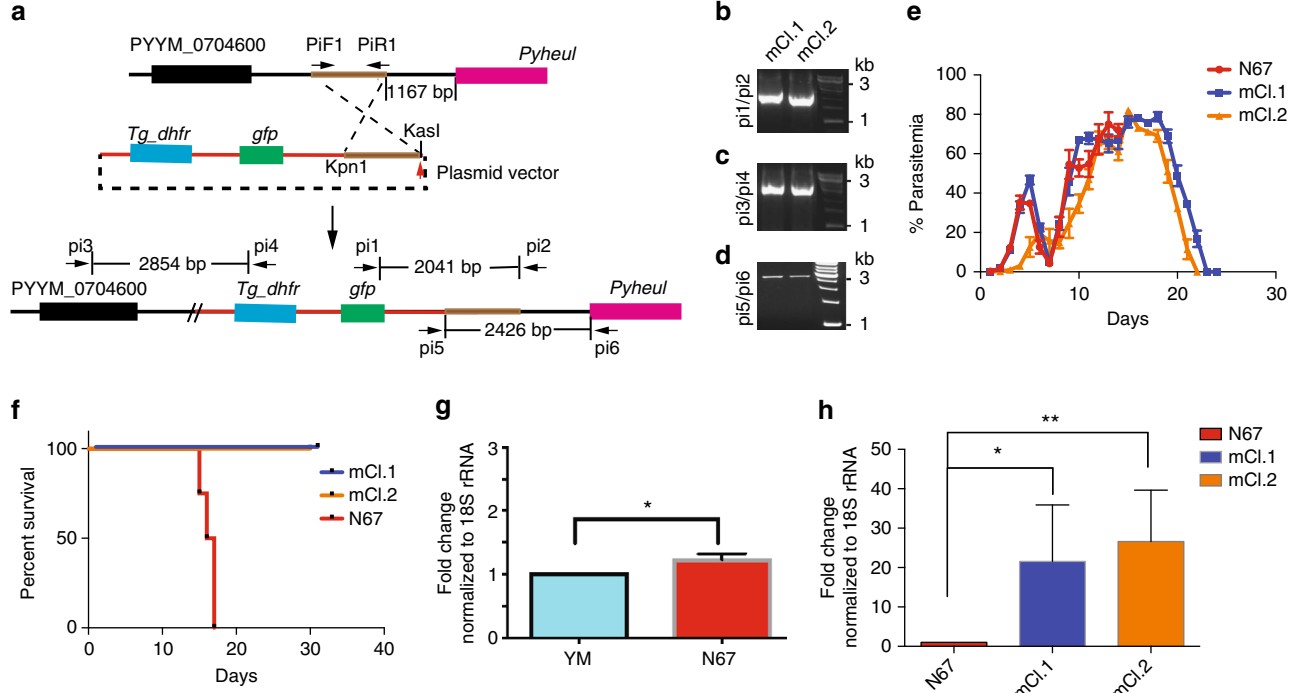

**Fig. 8** Alteration of *Pyheul* gene transcript level decreases host mortality. **a** *Diagrams* showing plasmid construct and strategy to insert a drug-resistant cassette into the 5′ untraslated region (5′ UTR) of the putative HECT E3 ubiquitin ligase (PyHEUL) gene. *Tg_dhfr*, *Toxoplasma gondii* dihydrofolate reductase; *gfp*, green fluoresecent protein gene. **b, c, d** PCR products amplified from genetically modfied clones using primers as indicated in (**a**). **e** Dynamics of the mean parasitenia of N67 and two mutants with insertion of a plasmid at 5′UTR of the *Pyheul* gene. SDs are from three to five infected mice each. **f** Mortality rates of mice infected with N67 and the two mutant clones. **g** *Pyheul* mRNA transcript levels of N67 and YM parasites, normalized to those of 18 S rRNA. The level of YM was treated as 1. **h** *Pyheul* mRNA transcript levels expressed as fold changes (compared to that of N67) in the mutant clones. The transcript levels were calculated results of RT-qPCR using primers specific for the PyHECT E3 ligase and normalization of expression levels to those of parasite 18 s rRNA. For (**g, h**), at least three indepepndent repeats. Two-sided *t*-test, $*P < 0.05$; $**P < 0.01$

parasitemia was reproducible in independent infections (Supplementary Table 6), suggesting relatively stable growth characteristics of the clones. The most likely explanation for the variations between the parasite clones is that additional changes, possibly caused by off-target effects of CRISPR/Cas9, occurred during transfection and drug selection, and the changes contributed to the unexpected growth phenotypes. To minimize the influence of host genetic background and experimental variations such as parasite injection, we mixed two parasites, N67-R1[N67-421] and N67-R1[17XL-234], at ~1:1 ratio and infected mice. We then measured the ratios of day-4 parasite DNA over day-2 DNA based on the peak heights of electropherograms to estimate parasite growth rates and showed that the N67 parasite having R1 segment replaced with that of 17XL (N67-R1[17XL-234]) increased its proportion from day 2 to day 4 (Fig. 6f and Supplementary Fig. 12), suggesting an increased growth rate after the replacement. Nonetheless, although the results from infections of cloned R1 exchanged parasites are inconclusive, the difficulty in cloning 17XL-R1[N67] parasites suggests that insertion of N67 R1 DNA into 17XL makes the parasite growth slower than the wild-type parasite.

To provide additional evidence to support the causative relationship of PyHEUL and parasite growth, we performed a second allelic exchange experiment replacing a 465 bp segment (from 188,788 to 189,252 bp) with five AA differences (R2) between 17XL and N67 (Fig. 7a, b). We were able to obtain self-replacement (N67-R2[N67] and 17XL-R2[17XL]) and allelic exchanged N67-R2[17XL] parasites, but not the17XL-R2[N67] parasite after five independent transfections and screening (Fig. 7c, d). The failure to obtain 17XL-R2[N67] parasite again suggests that

replacement of 17XL R2 segment with that of N67 significantly reduces parasite growth rate or viability, consistent with the difficulty in obtaining 17XL-R1[N67] parasite in the R1 replacement experiments. Infection of mice with 10 cloned R2 replacement parasites (N67-R2[17XL], including repeats of N67-R2[17XL-33] and N67-R2[17XL-52]) showed significantly higher average day-4 parasitemia of the N67-R2[17XL] parasites than those of the parental N67 or self-replaced N67-R2[N67] parasite (Fig. 7e and Supplementary Table 7). Although there were variations in parasitemia among the parasite clones, the differences in parasitemia between the N67-R2[17XL] groups mostly came from outlier mice in the groups, particularly for those infected with N67-R2[17XL-11] or N67-R2[17XL-23] that had a mouse having 29.5% and 24.7% parasitemia, respectively. The variations were likely due to technical difficulty in injecting mice. The results from R2 replacement were more consistent than those of R1 replacement, supporting a conclusion that replacement of N67 R2 segment by that of 17XL could significantly increase day-4 parasitemia.

**Alternation of *Pyheul* expression reduces host mortality**. We next inserted a plasmid containing genes encoding *Toxoplasma gondii* dihydrofolate reductase (TgDHFR) driven by the *Pbdhfr* promoter and a green fluorescent protein gene driven by *Pbef1a* promoter (pL0017 plasmid backbone from MR4) into the 5′ region of the *Pyheul* gene in N67 parasites to alter gene expression (Fig. 8a). After pyrimethamine selection and parasite cloning, we obtained eight clones from three independent transfections, two of which were selected for further functional characterization. Insertion of the plasmid into *Pyheul* 5′

untraslated region (UTR) was confirmed using PCR (Fig. 8b–d). Of the two mutant clones, mCl.1 (for mutant clone 1) grew similarly to N67 before day 7 p.i., whereas the other clone (mCl.2) had lower parasitemia than that of N67 before day 6 (Fig. 8e). However, mice infected with the two mutant clones showed increased survival rates with all the infected mice being able to clear parasites by day 20 p.i., whereas all the mice infected with N67 died before day 20 (Fig. 8e, f). To rule out the possibility that the introduced plasmid DNA mediated the reduced host mortality, we introduced the same plasmid into the N67 strain at the 18S rRNA site (Supplementary Fig. 13a). Two separate clones were obtained after PCR analysis and parasite cloning (Supplementary Fig. 13b, c). Mice infected with the two parasites died at the same time as those infected with N67 (Supplementary Fig. 13d, e), suggesting that the plasmid itself did not significantly interfere with host mortality.

To determine whether the inserted fragment affected the expression of the *Pyheul* gene, we measured the *Pyheul* transcript levels of the gene in the two engineered parasite clones in comparison with that of N67 using RT-qPCR. The results showed that N67 had significantly higher levels of *Pyheul* transcript than YM, and the two engineered clones had significantly higher levels of *Pyheul* transcript than N67 (Fig. 8g, h). The higher transcript levels of *Pyheul* in the engineered clones suggest that the plasmid insertion affected *Pyheul* transcription. We also attempted to insert the plasmid closer to the gene (95 bp to 1.2 kb upstream of ATG), but were unable to get stable lines even after 12 independent transfections. The inability to insert the fragment closer to the gene suggests that the native promoter as well as the gene is essential for the parasite.

## Discussion

In this study, we linked the *Pyheul* gene to parasite growth and host mortality after performing genetic crosses and linkage analysis. We attempted to disrupt the gene, but failed to obtain a viable *Pyheul* KO parasite clone using three different strategies. We next replaced two segments (R1 and R2) in the 7.3 kb mapped region. We were able to obtain parasites with R1 cross- and self-replacement parasites and showed that replacing the N67 R1 segment with that of 17XL (N67-R$^{17XL}$) could significantly increase parasite growth (day-4 parasitemia), but the 17XL-R1$^{N67}$ parasite did not significantly alter the parasite growth rate (compared with 17XL or 17XL-R1$^{17XL}$). For the R2 replacement, we were not able to obtain any 17XL parasite having R2 segment replaced with that of N67 (17XL-R2$^{N67}$) after many transfections and sequencing screenings of parasite mixtures grown out of drug selection, whereas it was very easy to obtain N67-R2$^{17XL}$ parasite clones. One possible explanation for the difficulties in obtaining 17XL-R1/R2$^{N67}$ parasites was that the 17XL-R1$^{N67}$ and 17XL-R2$^{N67}$ parasites grew slower than the parental 17XL or 17XL-R1$^{17XL}$ parasites and were overgrown (and missed) when parasite cloning was performed (the 17XL-R2$^{N67}$ could be lethal). On the other hand, the N67-R2$^{17XL}$ grew faster than N67 and became the dominant population after a few days of growth. These results are consistent with the fact that N67 grows slower than 17XL in the early phase of infection and suggest that introduction of 17XL *Pyheul* sequences into N67 genome made parasites grow faster and they became easier to clone, whereas introduction of N67 into 17XL rendered slow-growing parasites that were missed at the cloning stage. The discrepancies in cloning efficiency of the modified parasites provide strong evidence that PyHEUL play a role in parasite growth. However, the observation that some 17XL-R1$^{N67}$ clones did not have significantly lower day-4 parasitemia than those of 17XL-R1$^{17XL}$ or 17XL parasite was somehow contradictory to the observations of more difficulty in

obtaining the 17XL-R1$^{N67}$ than cloning other modified parasites. One potential explanation is that the 17XL-R1$^{N67}$ clones we eventually obtained were among the few fast growers emerging from parasites with additional compensatory changes in the genome. Further investigations are necessary to support this speculation.

Another sign of additional changes that might occur in some parasites during parasite transfection, selection, and cloning was the parasites with unexpected growth phenotypes. Among the genetically modified clones, we also obtained self-replacement clones (17XL-R1$^{17XL}$ and N67-R1$^{N67}$) that had day-4 parasitemia < 15%, while ~90 and ~45% parasitemia, respectively, were expected for these clones. Changes other than the targeted modifications in the genome could have occurred in the parasites during the transfection and drug selection, or off-target cleavages by the Cas9 enzyme could play a role[38] in changing the parasite growth characteristics. In addition to potential unknown changes during transfection of the R1 segment, technical variation in mouse injection and batches of mice used over a period of several months could be sources of phenotypic variation. Among groups of four or five mice, sometimes an outlier mouse had parasitemia quite different from the rest of the group (for example, N67-R2$^{17XL}$-11 and N67-R2$^{17XL}$-23). Despite the variations in mouse injection and potentially unknown changes in the R1 replacement leading to less than perfect changes in growth phenotype after allelic exchanges, the difficulties in cloning both 17XL-R1$^{N67}$ and 17XL-R2$^{N67}$ parasites and the general patterns of faster growth of N67-R2$^{17XL}$ parasite clones (compared with N67 or N67 self-replacement) support a conclusion that the 17XL allele of *Pyheul* makes parasites grow faster and that *Pyheul* contributes to parasite growth in C57BL/6 mice.

There are different classes of ubiquitin ligases, and the ones that have a domain similar to the carboxy terminus of the E6-associated protein (~ 350 AAs) are called Homology to E6-associated protein Carboxy Terminus or HECT E3 ligases[39]. Ubiquitination of proteins has also been shown to regulate a wide range of cellular activities like protein degradation, molecular signaling, DNA repair, cell cycle control, transcription, endocytosis, intracellular trafficking, and immune responses[33, 40]. The ability of the *Plasmodium* parasites to ubiquitinate proteins during red blood stages has been reported[41], suggesting the presence of active ubiquitination pathways in *Plasmodium* parasites. The *P. falciparum* has 114 genes encoding ubiquitin-proteasome system-related proteins, including 54 E3 or E3-like ligases[40, 42]. Of the four HEULs identified in the *P. falciparum* genome, three are homologues to TOM1, UFD4, and HUL5, respectively, in the budding yeast *Saccharomyces cerevisiae*. The fourth one is similar to another ubiquitin ligase identified in *Arabidopsis*[40]. The PyHEUL (MAL8P1.23, PYYM_0704700) identified in this study is homologous to the TOM1 ligase with a conserved HECT domain at the C terminus and two domains of unknown function at the N terminus. The TOM1 enzyme in *S. cerevisiae* has recently been reported to be involved in cell cycle arrest after DNA damage, mediating CDC6 ubiquitination[43]. Interestingly, a yeast two-hybrid study of genome-wide *P. falciparum* protein interactions detected 27 proteins interacting with the PyHEUL homolog (PF3D7_0930300) of *P. falciparum*. These proteins included merozoite surface protein 1 (MSP1, PF3D7_0930300), cytoadherence-linked asexual gene 2 (CLAG2, PF3D7_0220800), protein with DNAJ domain (PF3D7_0532400), and chromodomain–helicase–DNA-binding protein 1 homolog (PF3D7_1023900)[44]. MSP1 is a protein localized on the merozoite surface, may play a role in parasite invasion or egress, and is recognized by the host immune system[45–47]. The PyHEUL could mediate MSP1 degradation, leading to reduced MSP1 protein levels. The PyHEUL may indirectly affect host immune

response and/or parasite invasion through regulating the levels of proteins such as MSP1 and/or CLAG2. Various E3 ligases have been shown to play a role in the modification of host immune signaling pathways in bacterial infections, facilitating the survival of the pathogens[48, 49]. In addition, a *P. falciparum* HEUL was associated with differential susceptibility to quinine drugs[50], and a mutated de-ubiquitinating enzyme was linked to increased resistance to chloroquine in *P. chabaudi*[51]. These observations point to important roles of ubiquitination processes in parasite responses to external pressures and in parasite survival, possibly by affecting the stability of proteins playing a role in parasite responses to drug and/or immune pressure. Although we do not have data on how PyHEUL regulates protein levels and host immune response, our preliminary observations provide an interesting hypothesis for further investigation.

Parasitemia, cytokine level, and host mortality are complex but related disease phenotypes, likely involving many parasite and host genes, which is consistent with our observation of multiple linked-parasite loci (Chr1, Chr7, Chr8 loci, and possibly others). A disease phenotype is also dependent on interactions of different parasites and/or host strains or species[5]. For an individual gene, both SNPs and indels in protein-coding region and differences in gene expression between parasites can contribute to the variation in disease phenotype. There are 97 nonsynonymous SNPs between YM/17XL and N67 (Supplementary Table 4), and the *Pyhuel* transcript level appeared to be significantly higher in N67 than in YM (Fig. 8g), suggesting that both AA substitutions and gene expression may contribute the phenotypic differences between N67 and YM/17XL. The results from our allelic exchange experiments and gene expression analyses suggest that both differences in the *Pyheul* gene sequence and expression can influence disease phenotypes, in addition to many unknown genes with differences between the parasites.

Results from our mixed infection experiments support that parasite growth, host immune response, and mortality are closely related phenotypes. The ability to invade both young and mature RBCs efficiently has been the explanation for the fast-growing and lethal phenotype in YM infection. Our results of mixed infection of YM and N67 showed that immune responses (higher levels of IFN-γ, IL-β, IP-10, MCP-1, MIG, etc.) stimulated by N67 parasites could suppress YM growth. A strong early IFN-I response, mediated by the MDA5/MAVS pathway, and increased numbers of activated macrophages in the spleen have been associated with the decline of N67 parasitemia in C57BL/6 mice[5]. These results also indicate that a strong host immune response stimulated by a specific parasite strain or subspecies (N67) can also affect the growth of a genetically diverse parasite (YM). Alternatively, the results may suggest that the YM parasite has mechanisms to inhibit some pathways of the host immune responses, allowing its rapid growth. Understanding the molecular basis of the cross-subspecies (species) inhibition may be important for designing vaccines against a diverse parasite population.

The locus on Chr1 significantly linked to day-7 parasitemia contained genes encoding two YIR and three fam-a proteins. Many YIR proteins are expressed on the surface of erythrocytes infected with late-stage asexual parasites[52]. In addition, the transcriptional profiles of the *yir* genes change rapidly and can be modulated by the host immune response[52, 53], suggesting interaction of the YIR proteins with the host immune system. Similarly, the fam-a proteins are recognized by host immune responses[54], and some may play a role in parasite binding to RBCs[55]. These observations support the possibility that some of the YIR or fam-a proteins play a role in modulating host immune responses to parasites; however, whether the genes in the Chr1 locus can modulate host immune responses requires further functional investigations.

In summary, our study has identified a component of the parasite ubiquitination pathway that may regulate parasite invasion efficiency and/or host immune responses. In addition to genetic mapping linking the *Pyheul* gene to parasite growth, we also obtained several lines of evidence to support the conclusion that PyHEUL plays a role in parasite growth and/or virulence: (1) the difficulty in obtaining 17XL parasites with replacements of N67 R1 and R2 DNA segments suggest that N67 DNA contributes to slower early parasite growth, consistent with the slow-growing phenotype of N67; (2) results from multiple N67 clones inserted with 17XL R2 segment (N67-R2[17XL]) significantly increased day-4 parasitemia; (3) changes in gene expression after plasmid insertion into *Pyheul* 5′ UTR alter parasite virulence. Additional studies are necessary to completely understand the molecular mechanism of the host–parasite interaction, including measurement of cytokine levels in the host and of changes in ubiquitination of parasite proteins in the *Pyheul*-modified parasite clones. Understanding the nature of the interactions involved in the modification of host responses and the identification of critical parasitic components involved will be important for developing better therapeutic approaches against malaria.

## Methods

**Parasites, mice, and measurement of parasitemia**. The origins of the two *P. yoelii* lines used in this study were described previously[28]. C57BL/6 female mice, 5–8-week-old, from Charles River Laboratories were injected as described[28, 30]. Briefly, an inoculum containing $1 \times 10^6$ of iRBC suspended in 100 μl of sterile phosphate-buffered saline, pH 7.4, from donor mice was injected intravenously into experimental C57BL/6 mice. Naive mice receiving an equivalent number of uninfected RBCs served as a negative control group. For infecting a donor mouse, a vial containing frozen iRBCs was thawed at room temperature, and 100 μl of the thawed iRBCs were injected into the peritoneum of the mouse. For allelic exchange experiments, female BALB/c mice, 5–8-week-old, were obtained from Shanghai Laboratory Animal Center, CAS (SLACCAS). Parasitemias were counted from Giemsa-stained thin blood smears using a light microscope. All animal procedures were performed in accordance with the approved protocols (#LMVR11E) by Institutional Animal Care and Use Committees at the National Institute of Allergy and Infectious Diseases (NIAID) DIR ACUC following the guidelines of the Public Health Service Policy on Humane Care and Use of Laboratory Animals and AAALAC and protocols approved by the Laboratory Animal Management and Ethics Committee of Xiamen University (permit #XMULAC20150080). Mice were randomly divided into groups for all experiments, and the groups were not blinded during parasitemia counting.

**Oocyst count of infected mosquitoes**. Infected mice 4 days after injection of iRBCs were fed to 70–100 *Anopheles stephensi* (Nijmegen strain) mosquitoes for 20 min. Uninfected mosquitoes were removed immediately post feeding by examining the blood-filled abdomen. Mosquito midguts were dissected 9–10 days post feeding, and oocysts were stained with 0.05% mercurochrome solution for 5 min. Stained oocysts were counted under ×10 magnification with a light microscope.

**P. yoelii genetic cross**. Experimental procedures for genetically crossing *P. yoelii* parasites have been described in detail previously[32]. Briefly, a C57BL/6 donor mouse was co-infected with both strains of the parasite in a desired parental ratio. This ratio was determined based on the relative ability of the individual strain to produce infective oocysts in the midgut of an infected mosquito[31]. Because the N67 strain produced 9–10 times more oocysts than the YM strain (Supplementary Fig. 2), a 9 (YM):1(N67) ratio in the number of parasites was used in the cross experiments. The donor mouse day-4 p.i. was exposed to mosquitoes for 20 min Sporozoites were collected from the mosquitoes 21 days post feeding and were injected intraperitoneally into a single mouse that was examined for the presence of erythrocytic stages daily. Parasites from the positive blood were diluted to 1 parasite per 100 μl blood in PBS and were injected into 100–120 individual mice each. The mice were examined for parasites daily for 7–9 days p.i., and blood samples from infected mice were collected for DNA extraction and for parasite preservation in liquid N2. The collected blood samples were passed through a Plasmodipur filter (Plasmodipur, the Netherlands) to remove white blood cells, and genomic DNA from the infected RBC pellet was isolated using the High Scale PCR template purification kit from Roche Diagnostics (Cat no:11796828001, Roche Diagnostics, USA).

**Genotyping of the recombinant progeny**. Cloned parasites were initially typed with 10 microsatellites to screen for potential recombinant progeny as described[36] using primers in Supplementary Table 2. DNA labeling and hybridization to a custom-made SNP-typing microarray were conducted essentially as described

using 1 μg of gDNA[27]. Scanned images were gridded and processed using NimbleGen v2.5 (Roche NimbleGen Inc.) to extract signal data. SNP genotypes were called when the probe sets at each locus indicated complementary SNP genotypes on both DNA strands. Parental clones were also hybridized in duplicate, and genotypes were scored at each locus as inheriting one of the parental genotypes or as N/A if a non-parental genotype was called. A total of 60 progeny with unique microsatellite patterns were genotyped with the SNP array.

**Clustering genotypes and phenotypes**. For clustering multiple-SNP average genotype, a per-chromosome average genotype was first calculated by encoding the SNPs into a numeric scale (0 = YM genotype, 2 = N67 genotype, 1 = missing genotype call). A matrix consisting of the mean genotypes, 14 chromosomes by 45 parasites, was clustered using Ward's method in JMP software (SAS Institute, Cary, NC). The cluster order of parasites was extracted and used to organize the heatmap of average genotype.

For clustering parasites based on phenotype (parasitemia and mortality), a matrix of daily parasitemia levels for 43 progenies and two parental strains for 25 days was used as the basis for ordering the parasites according to similarities in time course of parasiteima. Missing values (post-mortem time points) were imputed with final measured parasitemia level before clustering using Ward's method in JMP software (SAS Institute). A sensored version of the parasitemia heatmap was created using the order of parasites generated in the clustering.

**QTL analysis**. QTL analysis was performed using R/qtl library in the R12.2.2 software[56] as described previously[28]. Standard interval mapping (EM method) was employed, and a 5% threshold ($P = 0.05$) was used as the significant cutoff value after 1000 permutations. Correlation of parasitemia with the day of death was performed using Graph Pad Prism software version 6.0 (Graph Pad software, La Jolla, CA, USA).

**Measuring blood cytokines and chemokines**. Approximately 75 μl of blood was collected every other day from the tail vein into heparinized tubes and centrifuged at 13,000 r.p.m. for 15 min. The plasma was aliquoted and stored at −80 °C for further analysis. The levels of cytokines and chemokines were measured using a mouse magnetic 20-plex Bead Array kit (Invitrogen) in a Luminex-200 machine according to the manufacturer's instructions.

**RNA isolation and RT-qPCR**. RNA was isolated using an RNeasy mini kit following the manufacturer's protocol (Cat No./ID: 74104, Qiagen). Total RNAs were isolated from blood cells for real-time quantitative PCR (RT-qPCR) analysis. Infected mice were terminally bled on day-4 p.i., and the blood was collected in heparinized tubes. RNA was quantified using a Nanodrop spectrophotometer. Approximately 1 μg of RNA sample was first treated with DNase to remove any contaminating genomic DNA before cDNA synthesis using a QIAGEN reverse transcriptase kit. The cDNA sample was tested for the absence of genomic DNA contamination using tubulin alpha primers that gives different amplicon sizes with genomic DNA and cDNA templates (Supplementary Table 5). Two microliters (~ 100 ng) of the cDNA was used for RT-qPCR using primers specific for HECT E3 ligase. Products of parasite 18S rRNA were used for normalization of expression levels in the two strains.

**Plasmid constructs and parasite transfection**. For insertion at 5′ UTR, an 800 bp of genomic segment 1127 bp upstream of the HEUL open reading frame was amplified using the primers PiF1and PiR1 (Supplementary Table 5). The amplified fragment was cloned into the pL0017 plasmid vector (BEI Resources, NIAID, NIH, catalog #MRA-786, contributed by Andrew P. Waters) using KpnI and KasI restriction enzymes (Fig. 8a). The plasmid construct was linearized using KasI before parasite transformation that was performed according to methods described previously[36, 57]. Primers for making a construct for insertion into the 18S rRNA locus as control are also listed in Supplementary Table 5.

Knocking out of the HEUL locus was attempted using a linear construct according to the procedure described previously[35]. Briefly, an 825 bp 5′ UTR of the HEUL gene was amplified using primers PlF1 and PlR1, and a 900 bp of 3′ UTR of the HEUL gene was amplified using primers PlF3 and PlR3 (Supplementary Fig. 9 and Supplementary Table 5). A PCR fragment containing the hdhfr cassette was amplified from pL0007 vector deposited by AP Walters (https://www.mr4.org) using primers PlF2 and PlR2. Finally, a linear product was amplified using primers PlF1 and PlR3 and the three PCR fragments as DNA templates. Ten micrograms of the purified PCR product were used in transfection experiments. Additional primers were synthesized for detection of integration (Supplementary Fig. 9 and Supplementary Table 5).

**CRISPR/cas9 mediated genetic modification of the Pyheul gene**. The methods used in gene disruption and allelic exchanges were basically as described[37]. To construct the plasmid to completely delete the coding region of the Pyheul gene, we used primer sets 1K55/1K53 and 1K35/1K33 (Supplementary Table 5) to amplify 5′ UTR- and 3′ UTR regions as left and right homologous arms and cloned them into restriction sites of HindIII/NcoI and XhoI/EcoRI in the pYC vector,

respectively. For the plasmid to delete partial coding region (from 184,158 to 186,530) of the Pyheul gene, we amplified two DNA segments as left and right arms using primers sets 2K55/2K53 and 2K35/2K33 (Supplementary Table 5).

To construct plasmids for allelic replacement, we amplified two fragments (R1, from184,353 to 185,394 bp and R2, from 188,788 to 189,252 bp) and inserted the segments into the NcoI/XhoI sites of the pYC vector, respectively. To prevent binding of sgRNA to the target site in the donor template after integration, three synonymous nucleotide substitutions were introduced into the sgRNA-binding sites in the donor templates using synthetic oligonucleotides and PCR amplification. The sequences of the oligonucleotides and PCR primers for these experiments are list in Supplementary Table 5. Transfection, selection of transformed parasites, and parasite cloning were performed as described previously[37]. Integration events with introduced sequences in parasite mixtures were first detected using PCR amplification and DNA sequencing. Parasite cloning was followed if the introduced DNA sequences were detected; additional transfection and cloning were performed if no detectable integration event was found.

**Statistical methods**. Paired or unpaired t-test (two-sided) was used for all statistical analyses of the data samples in the current study.

**Data availability**. The authors declare that the data supporting the findings of this study are available within the article and its Supplementary Information files.

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

## Acknowledgements

This work was supported by the National Natural Science Foundation of China (#81572017 and #81220108019), by Fundamental Research Funds for the Central Universities (20720160057), by Project 111 of the State Bureau of Foreign Experts and Ministry of Education of China (B06016), and by the Division of Intramural Research, NIAID, National Institutes of Health, USA. We thank Cindy Clark of NIH library for editing.

## Author contributions

S.C.N., R.X., S.P., J.W., Y.Q., and R.T.E.: performed genetic crosses, gene knockout, allelic exchange, and other experiments; M.Z., S.G., and V.N.: bioinformatics and sequence analyses; M.S.O.: histochemistry; T.G.M. and J.T.: microarray and genotype/phenotype clustering; S.C.N.: data analysis and writing; C.A.L.: contributed reagents and wrote paper; S.L.: supervised experiments; J.L.: supervised and designed experiments, analyzed data, and wrote paper; X.-z.S.: conceived the project, designed experiments, analyzed data, and wrote paper.

## Additional information

**Competing interests:** The authors declare no competing financial interests.

