## [Peer review file · Nature Communications]

Reviewers' comments:

Reviewer #1

Expert in Plasmodium genetics

(Remarks to the Author):

Nair, Su, and colleagues describe a genetic cross undertaken in the *Plasmodium yoelii* rodent parasite to map the loci responsible for a difference in virulence between the YM and N67 lines. They find a strong signal on chromosome 7 by linkage analysis after cloning 43 recombinant progeny, and fine map the signal a Hect E3 ubiquitin ligase. They attempt to discover the mechanism by which this HEUL influences parasitemia and host mortality and discover some clues, but ultimately do not precisely elucidate the mechanism.

This is a significant piece of work that leads to an interesting finding, and the described analyses appear to be appropriate and well executed. The manuscript could be improved by attention to some small suggested improvements:

Abstract, elsewhere: What does 'improved' host mortality mean? Would be clearer to say increased or decreased.

Given that the HEUL affects MSP1 expression, why is so much weight given to the idea that virulence differences are effected by modulation of host immune response? Is it not sufficient to hypothesize that reduced expression or abundance of MSP1 could directly reduce parasite growth and virulence through inhibition of erythrocyte invasion?

How can direct or indirect mechanisms be in accord with the observation that infections with mixtures of strains show intermediate levels of parasitemia/virulence? I.e., how would reduced expression of MSP1 in N67 reduce host immune response to a YM co-infection with higher MSP1 expression?

I feel this manuscript is a significant body of work that should be published even without a definitive understanding of mechanism, but these points were confusing in the Discussion. Clarification of the outstanding questions would be a helpful improvement.

Reviewer #2

Expert in Plasmodium genetics

(Remarks to the Author):

This manuscript describes a project that maps malarial loci involved in eliciting different host responses in mice to a murine malaria. Different substrains of malaria elicit very different host responses. The authors choose two related *P. yoelii* parasite clones (YM and N67) that both result in death by the murine host (C57/BL6). However the kinetics of parasitemia are very different between the two parasite clones with the YM strain causing death in 6-7 days and the N67 requiring at least 16 days to kill the host. Interestingly the host response to co-infection by both strains is N67-type dominant with the kinetics of infection resembling that of the N67 clone. Cytokine analysis showed different responses to the two different clones and these followed the N67-type response in mixed infections.

Genetic crosses between N67 and YM resulted in 43 unique recombinant progeny. These were all used to infect mice and a range of infection and outcome kinetics were observed that traversed

that of the two parental clones and even produced non-lethal strains. Two loci with statistically significant LOD scores were seen on chromosomes 1 and 7. A non significant linkage was seen on chromosome 8. QTL analysis of parasitemias on various days demonstrated that the chromosome 7 locus was the more important of the two.

The authors then went on to identify genes within the two linked loci. They identified a locus on chromosome 7 which encompasses two "excluding" crossovers, leaving 7.3kb of "included" sequence that contains an E3 ubiquitin ligase. They were unable to knock this gene out but modified its expression by inserting a plasmid in the 5'untranslated region of the gene and were able to show that the phenotype changed, leading to a more benign infection in the mice.

This work is indeed original. It would be of very great interest if the results presented around the E3 ubiquitin ligase identification were more solid than presented in this manuscript. This would perhaps identify a potential drug or vaccine target. However in the absence of this, the actual mapping data is of no major interest in its own right and the potential interest in the differential growth kinetics has not been sufficiently well developed to be of interest per se.

The statistical analysis is reasonably straightforward and appears robust.

However there are certain elements in the description of the project that give pause for thought:

1. The two parasite clones, (YM and N67) are quite different genetically. Sufficiently different to be able to distinguish all chromosomes. This probably is more genetic distance than would enable them to be called "isogenic", a term usually used to identify lines with localised and limited genetic differences. It also means that there are possibly many loci involved in the studied phenotypes. These may give individual effects below the limit able to be detected by the small number of progeny (43) under study.

2. The localisation of the 7.3kb of chromosome 7 containing the E3 ubiquitin ligase gene as the causative gene is not convincing. This was localised based on two recombination events. However, as the authors acknowledge, this locus contributes to a quantitative trait. It is therefore not certain, as it would be if this were a Mendelian trait, that these flanking recombination events do indeed delineate a minimal functional region. There are obviously other regions of the genome that need to be taken into consideration. This could be done by purifying this locus as a congenic interval, however, in the absence of more work, these recombinations cannot be used as the definition of the minimal genetic interval in a complex trait.

3. There is discussion about the clusters of mutations in the E3 ubiquitin ligase gene as being due to immune selection. However later, it is stated that none of these mutations give rise to an amino acid change. This would tend to rule out an immune selection.

4. It was impossible to knock this gene out, indicating that it is essential for parasite survival and growth. However when a plasmid was inserted into the 5' region of the gene, expression of the ubiquitin ligase was drastically increased, by perhaps 20-30 fold and this was claimed as evidence that this gene mediated the difference in phenotype between N67 and YM parasites. However the difference in expression in Pyheul between these parasites is of the order of 20%, which is some 100-150 times less than the difference seen between N67 and the genetically modified parasites. It is therefore eminently plausible that this very large difference may be effecting outcome through some totally different mechanism. In fact the phenotype that is seen is survival of the mice. This means that the ability of the parasite to survive is compromised. Messing with any "essential" protein is likely to give rise to a parasite that is less fit and therefore more likely to give rise to a resistant host. Therefore this data does not convince this reviewer that Pyheul is the correct, causative gene.

5. Similarly, discussion of candidature of other genes is based on even more tenuous evidence and

should not be included.

6. For the readers to be completely convinced that this is the correct gene, an allelic substitution needs to be performed, or at least the production of animals congenic for minimal genetic region need to be developed. Both of these would necessarily rule out the candidature of neighbouring genes.

The references are adequate.

The paper contains typos, which could be corrected. It is clearly written, however, as delineated above, the conclusions are not well supported by the data.

Responses to reviewers' comments:

Reviewer #1

Nair, Su, and colleagues describe a genetic cross undertaken in the Plasmodium yoelii rodent parasite to map the loci responsible for a difference in virulence between the YM and N67 lines. They find a strong signal on chromosome 7 by linkage analysis after cloning 43 recombinant progeny, and fine map the signal a Hect E3 ubiquitin ligase. They attempt to discover the mechanism by which this HEUL influences parasitemia and host mortality and discover some clues, but ultimately do not precisely elucidate the mechanism.

This is a significant piece of work that leads to an interesting finding, and the described analyses appear to be appropriate and well executed. The manuscript could be improved by attention to some small suggested improvements:

Abstract, elsewhere: What does 'improved' host mortality mean? Would be clearer to say increased or decreased.

Response: Thank you for the nice comments. We have replaced “improved” with “decreased” or “increased”.

Given that the HEUL affects MSP1 expression, why is so much weight given to the idea that virulence differences are effected by modulation of host immune response? Is it not sufficient to hypothesize that reduced expression or abundance of MSP1 could directly reduce parasite growth and virulence through inhibition of erythrocyte invasion?

Response: Thank you for the suggestion. The mixed infection in figure 2 shows that host immune response to N67 plays a role in controlling YM growth, leading us to speculate on the influence of immunity on parasite growth. Indeed, it is possible that MSP1 can also affect parasite invasion directly. We have added: “Alternatively, reduced expression of MSP1 could directly affect parasite growth and virulence through reduced erythrocyte invasion efficiency” to the discussion (**line 527-528**).

How can direct or indirect mechanisms be in accord with the observation that infections with mixtures of strains show intermediate levels of parasitemia/virulence? Ie, how would reduced expression of MSP1 in N67 reduce host immune response to a YM co-infection with higher MSP1 expression?

Response: We believe that mice infected with N67 initiate an innate response that also can suppress the growth of YM strain, but the immune response may not be 100% effective in controlling YM growth. Infection with N67 is known to stimulate a strong type I interferon response that plays a role in suppressing parasitemia (Wu et al., PNAS, 2014, E511-20). Many parasite molecules can also influence this N67 specific response, and we still do not know the mechanism of how the response to N67 affects YM growth. It may be too early to conclude that reduced expression or abundance of MSP1 can

reduce parasite growth and virulence through immune responses. We have removed the MSP-1 data from the manuscript.

I feel this manuscript is a significant body of work that should be published even without a definitive understanding of mechanism, but these points were confusing in the Discussion. Clarification of the outstanding questions would be a helpful improvement.

Response: We have removed or reduced the discussion on immunity, MSP-1, and parasite growth, and hope these revisions improve the manuscript.

Reviewer #2

*This manuscript describes a project that maps malarial loci involved in eliciting different host responses in mice to a murine malaria. Different substrains of malaria elicit very different host responses. The authors choose two related *P. yoelii* parasite clones (YM and N67) that both result in death by the murine host (C57/BL6). However the kinetics of parasitemia are very different between the two parasite clones with the YM strain causing death in 6-7 days and the N67 requiring at least 16 days to kill the host. Interestingly the host response to co-infection by both strains is N67-type dominant with the kinetics of infection resembling that of the N67 clone. Cytokine analysis showed different responses to the two different clones and these followed the N67-type response in mixed infections.*

Genetic crosses between N67 and YM resulted in 43 unique recombinant progeny. These were all used to infect mice and a range of infection and outcome kinetics were observed that traversed that of the two parental clones and even produced non-lethal strains. Two loci with statistically significant LOD scores were seen on chromosomes 1 and 7. A non significant linkage was seen on chromosome 8. QTL analysis of parasitemias on various days demonstrated that the chromosome 7 locus was the more important of the two.

The authors then went on to identify genes within the two linked loci. They identified a locus on chromosome 7 which encompasses two "excluding" crossovers, leaving 7.3kb of "included" sequence that contains an E3 ubiquitin ligase. They were unable to knock this gene out but modified its expression by inserting a plasmid in the 5'untranslated region of the gene and were able to show that the phenotype changed, leading to a more benign infection in the mice.

This work is indeed original. It would be of very great interest if the results presented around the E3 ubiquitin ligase identification were more solid than presented in this manuscript. This would perhaps identify a potential drug or vaccine target. However in the absence of this, the actual mapping data is of no major interest in its own right and the potential interest in the differential growth kinetics has not been sufficiently well developed to be of interest per se.

The statistical analysis is reasonably straightforward and appears robust.

However there are certain elements in the description of the project that give pause for thought:

1. The two parasite clones, (YM and N67) are quite different genetically. Sufficiently different to be able to distinguish all chromosomes. This probably is more genetic distance than would enable them to be called "isogenic", a term usually used to identify lines with localised and limited genetic differences. It also means that there are possibly many loci involved in the studied phenotypes. These may give individual effects below the limit able to be detected by the small number of progeny (43) under study.

Response: The reviewer is correct that the two parasites are diverse, and it is not appropriate to call N67 and YM isogenic. The isogenic pairs are YM and 17XNL or N677 and N67C. N67 and YM are very different genome-wide.

We also agree with the reviewer that multiple genes (more than those in the mapped loci) likely play a role in the disease phenotypes. This is common for a disease trait or phenotype, and is one of the problems we have. No matter how we modify or replace a candidate gene, we are unlikely to be able to ‘restore’ a disease phenotype perfectly or change a parental phenotype into another. For drug resistance genes, replacement or modification of a candidate gene may restore a drug resistance phenotype because the effects of mutations are often dominant with strong penetrance. We now have a paragraph to discuss the influences of multigenes, multiple substitutions in a gene, and change in gene expression on disease phenotypes (**line 538-553**).

2. The localisation of the 7.3kb of chromosome 7 containing the E3 ubiquitin ligase gene as the causative gene is not convincing. This was localised based on two recombination events. However, as the authors acknowledge, this locus contributes to a quantitative trait. It is therefore not certain, as it would be if this were a Mendelian trait, that these flanking recombination events do indeed delineate a minimal functional region. There are obviously other regions of the genome that need to be taken into consideration. This could be done by purifying this locus as a congenic interval, however, in the absence of more work, these recombinations cannot be used as the definition of the minimal genetic interval in a complex trait.

Response: First, we agree with the reviewer that other regions in the genome play a role on the phenotypes, which makes it difficult to modify a gene and change the whole disease phenotype completely. One approach to “purifying this locus as a congenic interval” is to clone more progeny; however, it is become more and more difficult to obtain new independent progeny because we kept getting the same redundant clones due to asexual replication. We have 43 progeny from this cross, which already represents the largest number of progeny ever produced from a malarial cross as far as we know.

LOD scores decline from both sides of a causative gene as recombination occurs, which was the rationale for us to pick the candidate gene E3 ligase. The 7.3 kb within the E3

ligase has the highest LOD score, and the LOD scores declined from both sides as recombination occurs. We agree that a final functional verification such as gene knockout or allelic exchange would be an ideal approach to prove the causative gene. Unfortunately, as the reviewer correctly pointed out, the nature of multigene traits makes it difficult to change a phenotype completely after modifying a candidate gene.

We now have performed additional experiments to confirm that the E3 ligase is essential for parasite viability and to functionally show that it can influence parasite growth using the newly developed CRISPR/cas9 method. We used three strategies (two CRISPR/cas9 constructs and one traditional double crossover) to disrupt the gene, but failed to obtain any parasites with disrupted E3 ligase gene. We also attempted to exchange two polymorphic coding sequences in the 7.3 kb mapped region and provided evidence that replacements of the DNA segments could affect parasite growth. These data, including two figures, three supplementary figures (new Fig. 6 and 7; new Fig. S10-S12), and two new supplementary tables (Supplementary Table 6 and 7) are added to the revised manuscript.

3. There is discussion about the clusters of mutations in the E3 ubiquitin ligase gene as being due to immune selection. However later, it is stated that none of these mutations give rise to an amino acid change. This would tend to rule out an immune selection.

Response: Sorry for the confusion. There are a large number of amino acid changes (93) in the E3 ligase gene between the N67 and YM parasites. However, none of the nucleotide substitutions in the conserved C-terminal ligase domain (the last 1.3 kb) change the amino acid. We have re-worded the sentences to clear this point: “**There were two polymorphic microsatellites and 225 single nucleotide polymorphisms (SNPs), including 97 nonsynonymous SNPs, between YM and N67; however, no AA substitution was observed in the C-terminal HECT domain³² (~1.3 kb; Supplementary Table 4).**” (line 279-282). We also deleted the statement of immune selection.

4. It was impossible to knock this gene out, indicating that it is essential for parasite survival and growth. However when a plasmid was inserted into the 5' region of the gene, expression of the ubiquitin ligase was drastically increased, by perhaps 20-30 fold and this was claimed as evidence that this gene mediated the difference in phenotype between N67 and YM parasites. However the difference in expression in Pyheul between these parasites is of the order of 20%, which is some 100-150 times less than the difference seen between N67 and the genetically modified parasites. It is therefore eminently plausible that this very large difference may be effecting outcome through some totally different mechanism. In fact the phenotype that is seen is survival of the mice. This means that the ability of the parasite to survive is compromised. Messing with any "essential" protein is likely to give rise to a parasite that is less fit and therefore more likely to give rise to a resistant host. Therefore this data does not convince this reviewer that Pyheul is the correct, causative gene.

Response: Reviewer raised an interesting point, and we agree that the expression levels and phenotypic changes were not proportional. In addition to expression level difference,

there are also a large number of amino acid substitutions between the N67 and YM parasites that can contribute to the phenotypic differences. For the two mutants, the gene sequences are the same as that of N67; the only expected difference is the gene expression level (except unknown changes during transfection). Therefore, both amino acid substitution and gene expression can affect parasite growth/virulence between N67 and 17XL, whereas gene expression level difference is the main factor affecting the phenotypic differences between the two mutants and N67.

Nonetheless, the data provide preliminary evidence that changing the expression of this gene can also affect parasite survival and growth, even though the effects could be mediated through other genes as well. As discussed above, we have provided additional evidence to support that the E3 ligase can affect parasite growth after replacement of some amino acids. Additionally, we have removed the MSP1 data and toned down the discussion of the change of expression on the disease phenotypes.

5. Similarly, discussion of candidature of other genes is based on even more tenuous evidence and should not be included.

Response: We have removed the MSP1 data. We discuss the chromosome 1 locus to point out the multigene nature of the disease phenotypes.

6. For the readers to be completely convinced that this is the correct gene, an allelic substitution needs to be performed, or at least the production of animals congenic for minimal genetic region need to be developed. Both of these would necessarily rule out the candidature of neighbouring genes.

Response: We have provided two sets of allelic exchange data to further support our mapping data (New Fig. 6, Fig. 7, and supplementary Fig. 12, 13, and 14).

The references are adequate.

The paper contains typos, which could be corrected. It is clearly written, however, as delineated above, the conclusions are not well supported by the data.

Response: We have gone over the manuscript to fix the typos; hope we found them all.

Finally, we would like to thanks the reviewers for the constructive comments/suggestions that have helped greatly improve our manuscript.

REVIEWERS' COMMENTS:

Reviewer #2 (Remarks to the Author):

Thankyou for addressing my comments. I am comfortable that the changes you have made and the reply to my comments have been adequately addressed. I believe your claims are adequately backed by evidence. However I do agree with the other reviewer that there is a possibility that another gene is involved. However I think that the effort required to eliminate, what is a very small amount of doubt, is greater than the benefit of publishing the paper as it stands at the present.